

# A trait database for marine copepods
**Philipp Brun[1], Mark R. Payne[1] and Thomas Kiørboe[1]**
[1]{ Centre for Ocean Life, National Institute of Aquatic Resources, Technical University of
Denmark, DK-2920 Charlottenlund, Denmark }
Correspondence to: P. Brun (pgbr@aqua.dtu.dk)
## Abstract
The trait-based approach is gaining increasing popularity in marine plankton ecology but the
field urgently needs more and easier accessible trait data to advance. We compiled trait
information on marine pelagic copepods, a major group of zooplankton, from the published
literature and from experts, and organised the data into a structured database. We collected
9345 records for 14 functional traits. Particular attention was given to body size, feeding
mode, egg size, spawning strategy, respiration rate and myelination (presence of nerve
sheathing). Most records were reported on the species level, but some phylogenetically
conserved traits, such as myelination, were reported on higher taxonomic levels, allowing the
entire diversity of around 10 800 recognized marine copepod species to be covered with few
records. Besides myelination, data coverage was highest for spawning strategy and body size
while information was more limited for quantitative traits related to reproduction and
physiology. The database may be used to investigate relationships between traits, to produce
trait biogeographies, or to inform and validate trait-based marine ecosystem models. The data
can be downloaded from PANGAEA, doi:10.1594/PANGAEA.862968.
## Keywords
Marine copepods, zooplankton, traits, body size, egg size, feeding mode, spawning strategy,
respiration rate, myelination



## 1 Introduction

The trait-based approach is an increasingly popular framework in ecology that aims to describe the structure and function of communities or ecosystems in a simple way. It seeks to identify the main characteristics of organisms that control their fitness (Litchman et al., 2013). Organisms must be successful in three main missions in order to thrive: feeding, survival and reproduction. Functional traits determine the outcome of one or several of those missions.

Functional traits are heritable properties of the individual that are interrelated through trade-offs and selected by the environment. They are measurable on the individual without any assisting information (Violle et al., 2007). Swimming behavior, therefore, is a functional trait of some animals, but preferred habitat is not, as it depends on the characterization of the environment in which an individual occurs.

The trait-based approach is well established in plant ecology (e.g., van Bodegom et al., 2014; Westoby et al., 2002) and more recently also in marine plankton ecology (Barton et al., 2013; Litchman and Klausmeier, 2008; Litchman et al., 2013). One key group of marine zooplankton, for which traits and trade-offs are relatively well understood, is copepods (Kiørboe, 2011). These ubiquitous crustaceans typically dominate the biomass of zooplankton communities (Verity and Smetacek, 1996), play a central role in marine food webs, and affect the global carbon cycle (Jónasdóttir et al., 2015).

We focus here on a set of 14 commonly-described functional traits for marine copepods, for which data are available (Fig. 1). The set includes one trait affecting all life missions, three feeding-related, six growth-related and three reproduction-related traits. Body size affects all life missions as it is related to several essential properties including metabolism, feeding, growth, mortality, mobility, and prey size (Litchman et al., 2013). Feeding-related traits include clearance rate, i.e., the effective volume of water cleared for prey items per unit of time, when the prey concentration is low (Kiørboe and Hirst, 2014); maximum ingestion rate - the feeding rate at non-limiting food concentration (Kiørboe and Hirst, 2014); and feeding mode (behaviour) (Kiørboe, 2011). For the latter, the following behaviours are separated: Ambush feeding copepods remain largely immobile and wait for approaching prey. Cruise feeding copepods move actively through the water in search for prey. Feeding-current feeders produce a current by beating their appendages and capture entrapped prey. Particle feeding copepods colonize large aggregates of marine snow on which they feed for extended periods, and parasites colonize larger hosts, such as fish, from which they feed. Growth related traits



include maximum growth rate (the maximum amount of body mass gained per unit time), and
the number of generations per year. Reproductive traits include spawning strategy, which
distinguishes between free-spawners that release their eggs into the water, and sac-spawners
that carry their eggs until hatching, egg size, clutch size (eggs produced in one 'spawning
event'), and fecundity (the number of eggs produced over the life-time of a female). Finally,
the traits related to survival are myelination (the insulation of nerve tracts with membraneous
tissue, which greatly enhances the speed of signal transmission and allows rapid response to
predators (Lenz et al., 2000)), respiration rate, the volume of oxygen consumed per unit time,
hibernation, which allows individuals to endure adverse conditions over seasonal time frames,
and resting eggs, which can endure adverse conditions over several decades (Williams-
Howze, 1997).
Here, we followed a recent call for efforts to collect trait data (Barton et al., 2013), and
established a database for the 14 copepod traits introduced above. We screened the literature
for information on marine copepods, mainly on pelagic taxa. Particular attention was given to
the traits body size, feeding mode, egg size, spawning strategy, myelination, and respiration
rate, for some of which we have examined the biogeography elsewhere (Brun et al.,
*submitted*). We present data coverage as well as trait distributions for the most important
pelagic copepod families and discuss data collection methods as well as limitations. The data
can be found on PANGAEA: doi:10.1594/PANGAEA.862968.





## 2  Data

### 2.1  Origin of data

Our data consists primarily of material from previous data compilations on individual traits, complemented by information from the primary literature and expert judgements. In total 90 references were consulted, with a few sources contributing the majority of the data (Table 1). The primary literature was screened mainly for information on the focal traits of body size, feeding mode, egg size, spawning strategy, and respiration rate. For feeding mode, we also used expert judgement: feeding modes have been described in the literature only for a minor fraction of copepod species. Where no information on feeding mode was available, we studied the morphology of the feeding appendages and, if feasible, grouped the taxa into two categories of feeding activity (active versus passive feeding, see Sect. 2.2.1).

### 2.2  Trait information

Besides the ecological categorisation shown in Fig. 1, the traits considered may be separated as categorical/qualitative traits and continuous/quantitative traits, which involve different ways of data storage.

### 2.2.1  Qualitative traits

Here, qualitative traits include feeding mode, spawning strategy, myelination, hibernation, and resting eggs. We treat qualitative traits as unique either on the species level or on higher order taxonomic levels. For hibernation and resting eggs, we report records on the species level, including information about the observed life stage in the case of hibernation. Species for which hibernation and resting egg production has been observed may be considered as having the potential to express the trait, without necessarily expressing it in every individual.

Feeding mode, spawning strategy, and myelination were assumed to be conserved in the taxonomy, yet we are aware that this is not always the case (Sect. 4.2). Records are therefore reported also for genera, families and orders, assuming all species from the corresponding taxonomic branch carry the trait. We distinguish five not-necessarily-exclusive feeding modes, i.e., ambush feeding, particle feeding, feeding-current feeding, cruise feeding and parasitic feeding (Kiørboe, 2011). Feeding modes are further clustered into different feeding activity levels (Table 2). Spawning strategy distinguishes between free-spawner and sac-



spawner that may be separated further to 'single egg-sac', 'double egg-sac' or 'egg mass'.
Finally, myelination distinguishes between myelinated and amyelinated taxa.

### 2.2.2 Quantitative traits

Quantitative traits include three size traits, four physiological rate traits, fecundity and number
of generations per year. Where possible, we report mean, minimum, and maximum trait value
as well as standard deviation and sample size for each record. Quantitative traits were
collected mainly for adults, but where available we also include information on juvenile life
stages. Several records may exist for each species and life stage/sex, originating from
different measurements or references. In some cases quantitative traits are reported on
taxonomic levels higher than species. This is usually due to limited taxonomic resolution, and
therefore such records should not be assumed to represent the entire taxonomic branch. For
each quantitative trait, we defined standard units in which the data is reported. Where
conversions were not straight forward, we report different 'types' of trait measurements, e.g.,
we distinguish between 'total length' and 'prosome length' for body size and between 'outer
diameter' and 'μg carbon' for egg size. The taxonomic overview of quantitative traits shown
below is based on species-wise averages of the data, restricted to adult individuals where life-
stage matters.

### 2.3   Meta information

### 2.3.1 Taxonomy

Around 10 800 marine copepod species are currently recognised (Walter and Boxshall, 2016).
Taxonomic classification of these small crustaceans is not trivial and has changed
considerably over the past century. In order to ensure consistency, all the taxa reported were
updated based on the latest (June 2 2016) (re)classification by Walter and Boxshall (2016)
with the finest possible resolution on the species level. We also added the full taxonomy of
marine copepods to our data tables in order to allow easy translation of the records to the
desired taxonomic level. However, we encourage readers to use the online version on
www.marinespecies.org/copepoda instead, to ensure that the information used is up to date.
For simplicity, we restrict the data presentation in this paper to a subset of the taxonomy,
mainly containing families with important pelagic species (Appendix A).





### 2.3.2 Life form

Copepods undergo a complex life cycle including an egg stage, six naupliar and six copepodite stages that may show distinct traits. Furthermore, distinct differences between sexes are possible, for example, through sexual size-dimorphism (Hirst and Kiørboe, 2014). If necessary, we therefore included information about life stage and sex of an individual in a 'life form' column (Table 3). Some authors distinguish between sexes already in copepodite stages IV and V (e.g., Conway, 2006). We disregard this separation to optimize consistency among the different sources.

### 2.3.3 Location

Traits can vary considerably as a function of the geographical location, in particular if they are observed on organisms in the field. Information about the geographical location, however, is not readily available in traditional data compilations. Nevertheless, we reported information about location where it was available.

### 2.3.4 Other

Further meta-information includes temperature, body mass and general comments. Physiological rate traits (growth rate, respiration rate, clearance rate and ingestion rate) depend on both body mass and temperature (Kiørboe and Hirst, 2014), which we also report for records of these traits. For body mass, we further distinguish 'dry mass' or 'carbon mass'. Further relevant meta-information may be provided in the 'Comment' field.

### 2.4 Data conversions

We consider our database to be primarily a source of information, and generally leave it up to the user to select methods and assumptions for aggregation and conversions, with the notable exception of physiological rate traits and egg size. Physiological rate traits largely stem from Kiørboe and Hirst (2014), who converted traits to carbon-specific values and to a standard temperature of 15 °C. For growth rate, clearance rate, and ingestion rate we included these converted values, while we recalculated them for respiration rate. We converted weight information to carbon content based on the empirical relationships provided in Kiørboe (2013). Temperature corrections were performed based on an empirical estimate of the $Q_{10}$ value from our data. The $Q_{10}$ value is the factor by which respiration rates increase when temperature is increased by 10°C and was estimated to be 2.14 (Appendix B). Egg size was



reported in part as carbon content. For comparability, we also report conversions of these
values to outer diameters assuming a spherical egg shape and a carbon density of $0.14 \times 10^{-6}$
$\mu$g C $\mu$m$^{-3}$ (Kiørboe and Sabatini, 1995).

## 3 Results

### 3.1 Data coverage

In total, the data tables include 9345 records for the 14 traits investigated. With 7131 records, by far the most information was available for body size (Fig. 2). However, for taxonomically clustered traits like myelination, only few records were necessary to cover all marine copepods. Similarly, relatively few records were available for hibernation and resting eggs, but they likely cover the existing information in the literature, and therefore the dominant species expressing these traits. For quantitative traits related to reproduction and physiology, information was generally more limited. Among taxa, the best data coverage was available for the order *Calanoida*. But also some non-calanoid families showed a relatively high data coverage, including *Oithonidae*, and *Oncaeidae*. For non-pelagic copepods, information was mainly available on myelination, and – for *Siphonostomatioida* – on feeding mode.

### 3.2 Body length

Total body length varies between 0.095 mm for *Acartia bacorehuiensis* and 17.4 mm for *Bathycalanus sverdrupi*, and is largest on average for calanoid copepods. Our data indicate shortest body lengths for the harpacticoid families *Harpacticidae*, *Discoidae*, and *Euterpinidae*, as well as for *Oithonidae* and *Oncaeidae*, with median total lengths of adults between 0.5 and 0.6 mm (Fig. 3a). Families with largest species are *Megacalanidae* followed by *Euchaetidae* and *Eucalanidae*, with median adult body lengths of 12.25, 6.51 and 5.54 mm, respectively. The highest interquartile range of body lengths is found for *Lucicutiidae* with 4.57 mm.

Body size does not only vary between species, but also within them. Not surprisingly body size increases considerably throughout the ontogeny of copepods (Fig. 3b). But significant variations in body size are also observed as a function of the geographic location. When compared in space, the prosome lengths of adult females of *C. finmarchicus* vary between about 2.5 and 3 mm across the North Atlantic, corresponding to a mass difference of a factor of over 1.7 (Fig. 3c).



### 3.3 Egg size


Egg diameter varies between 37.3 μm for *Oncaea media* and 870 μm for *Paraeuchaeta*
*hansenii*. The non-calanoid families covered (*Oncaeidae*, *Corycaeidae*, *Oithonidae*, and
*Euterpinidae*) tend to have smaller eggs than the calanoid families (Fig. 6a). With a median
diameter of 51.5 μm *Oncaeidae* is the family with the smallest egg sizes, while *Augaptilidae*
have the largest eggs with a median diameter of 554.3 μm. The highest diversity of egg
diameters is found for *Euchaetidae* with an interquartile range of 365.5 μm.

### 3.4 Myelination


Myelination only occurs in calanoid copepods and is assumed to be either consistently present
or absent within families. Major families with myelinated axons are *Aetideae*, *Calanidae*,
*Euchaetidae*, *Paracalanidae*, *Phaennidae*, and *Scolecitrichidae* (Fig. 7a).

### 3.5 Clearance rate


For adult copepods, carbon specific clearance rate corrected to 15 °C varies between 224 ml
h$^{-1}$ mg C$^{-1}$ for *Calanus pacificus* and 3067 ml h$^{-1}$ mg C$^{-1}$ for *Oithona nana*. On the family
level *Calanidae* show the lowest corrected clearance rates, whereas highest rates are found for
*Acartiidae* (Fig. 4a). The number of data points for adult copepods is only 18 for clearance
rate, as life stage information is missing for most records (Fig. 4b).

### 3.6 Ingestion rate


Carbon specific ingestion rate at 15 °C ranges between 15 μg C h$^{-1}$ mg C$^{-1}$ for *Calanus*
*pacificus* and 116 μg C h$^{-1}$ mg C$^{-1}$ for *Euterpina acutifrons*, when comparing adult
individuals. On the family level, lowest ingestion rates are found *Tortanidae*, and highest
values are found for *Euterpinidae* (Fig. 4c). Again, only 21 data points are available for
ingestion rates of adult copepods, as life stage information was missing for most records (Fig.
4d).

### 3.7 Growth rate


Specific growth rate at 15°C varies between 5 μg C h$^{-1}$ mg C$^{-1}$ for *Labidocera euchaeta* and
19 μg C h$^{-1}$ mg C$^{-1}$ for *Calanus finmarchicus*. In accordance, the families of these taxa,
*Pontellidae* and *Calanidae* have, respectively, the lowest and highest specific growth rates





among all families for which we have data (Fig. 4e). The highest diversity of growth rates is
also found for *Calanidae*, with an interquartile range of 10 µg C h$^{-1}$ mg C$^{-1}$.
**3.8  Respiration rate**
Specific respiration rate at reference temperature is lowest for *Hemirhabdus grimaldii* at 0.2
µL O$_2$ h$^{-1}$ mg C$^{-1}$ and highest for *Acartia spinicauda* at 79.5 µL O$_2$ h$^{-1}$ mg C$^{-1}$. Among
families, respiration rates are lowest for *Heterorhabdidae* (median = 0.5 µL O$_2$ h$^{-1}$ mg C$^{-1}$)
and highest for *Sapphirinidae* (median = 37.5 µL O$_2$ h$^{-1}$ mg C$^{-1}$) (Fig. 4f). The highest
interquartile range of specific respiration rates is found for *Acartiidae*. Most of the records on
respiration rates contain life stage information and are made for adult individuals (Fig. 4g).
**3.9  Feeding mode**
Feeding modes differ among taxonomic orders (Fig. 5). Calanoid copepods are active feeders,
and in some cases mixed feeders (*Acartiidae* and *Centropagidae*). Active feeding is also seen
in the order *Monstrilloida* and in the family *Oncaeidae* of the order *Poecilostomatoida*.
Passive feeding prevails in the orders *Cyclopoida* and some families of the order
*Harpacticoida*, as well as in the family *Corycaeidae* of the order *Poecilostomatoida*. Parasitic
copepods are found in the order *Siphonostomatoida* and in the family *Sapphirinidae* of the
order *Poecilostomatoida*.
**3.10 Generations**
The annual number of generations varies between 0.5 for *Calanus hyperboreus* and 9 for
*Acartia omorii*. On the family level *Eucalanidae* show the slowest life cycle with a median of
0.75 generations per year, while the median for *Centropagidae* is highest with 5.8 generations
per year (Fig. 6b).
**3.11 Clutch size**
Clutch size is below 35 for all taxa assessed, except for *Heterorhabdus norvegicus* from the
family *Heterorhabdidae*, for which it is 94 (Fig. 6c). Lowest clutch sizes are fond for
*Scaphocalanus magnus* (*Scolecitrichidae*) and *Tharybis groenlandica* (*Tharybidae*), with 1.6
and 2, respectively.





### 3.12 Fecundity

Fecundity ranges from 113 for *Pseudodiaptomus pelagicus* to 2531 for *Sinocalanus tenellus* (Fig. 6d). The largest interquartile range of fecundity is observed for *Centropagidae*.

### 3.13 Spawning strategy

Free spawning is only reported for calanoid copepods (Fig. 7b). In most cases spawning strategy is assumed to be conserved within family with the exception of *Aetideae*, *Arietellidae*, *Augaptilidae,* and *Clausocalanidae*. Important free spawning families are *Acartiidae, Calanidae*, *Paracalanidae*, *Phaennidae*, *Pontellidae* and *Scolecitrichidae*.

### 3.14 Hibernation

We found literature reports on hibernation for 28 species, mostly belonging to the family *Calanidae* (Fig. 7c). Further families with hibernating species are *Acartiidae*, *Clausocalanidae*, *Eucalanidae*, *Metridinidae*, *Pontellidae*, *Rhincalanidae*, *Stephidae*, and *Subeucalanidae*.

### 3.15 Resting eggs

The capacity to produce resting eggs has been observed for 47 species in total. Most of these species belong to the families *Acartiidae*, and *Pontellidae* (Fig. 7d). Further families with resting egg producing species are *Centropagidae*, *Sulcanidae*, *Temoridae*, and *Tortanidae*.



## 4  Discussion


We collected information on more than a dozen functional traits of marine copepods, and
combined it into a structured database. Our work complements recent and ongoing efforts to
develop zooplankton trait data collections. As for the collection of Benedetti *et al.* (2015), we
focused on those traits of marine copepods that are the main determinants of fitness, also
referred to as response traits (Violle et al., 2007). However, our collection covered the global
ocean rather than the Mediterranean Sea and a different, though overlapping, set of traits.
Hébert *et al.* (2016) recently published a trait database on marine and freshwater crustacean
zooplankton, which complementarily focuses on effect traits - traits which are expected to
impact aquatic ecosystems. Besides a few overlapping traits, this database mainly contains
information about body composition and excretion rates. Another noteworthy, ongoing effort
is the website maintained by Razouls *et al.* (2005-2016), who provide an impressive
collection of information for around 2600 marine pelagic copepod species. While they focus
on morphological descriptions, they also provide body length information, which in an
aggregated way was also included in this database. In terms of taxonomic breadth and
coverage of key functional traits as defined by the framework of Litchman *et al.* (2013) (Fig.
1), however, the data collection presented here is likely the most extensive. Nevertheless, our
database has several limitations which should be considered.

### 4.1  Trait definitions


There are uncertainties regarding the definition of some traits and their associated trade-offs,
in particular for hibernation and feeding mode. While we treat hibernation as a discrete
phenomenon, in reality a host of hibernation forms exist, differing considerably in the degree
to which metabolism is reduced (Ohman et al., 1998). Similarly, there are several feeding
mode classifications in the literature. We defined feeding modes after (Kiørboe, 2011), using
trade-offs in feeding efficiency and predation risk as classification criteria. We note that the
separation between cruise and feeding-current feeding is gradual, and that many species are
intermediate between these two categories. This is why we collectively categorize these
feeding modes as active, which is distinctly different from passive ambush feeding.
Other classification schemes differ in particular with respect to ambush feeding. We define
ambush feeding as a passive sit-and-wait feeding mode that targets motile prey with raptorial
prey capture, which applies primarily to *Oithona* and related taxa. Alternatively, ambush





feeding is sometimes defined solely based on raptorial prey capture (e.g., Benedetti et al.,
2015; Ohtsuka and Onbé, 1991), but raptorial prey capture can also be observed in cruise and
feeding-current feeders. Feeding types are sometimes also classified based on diet, e.g.,
herbivorous, carnivorous, or omnivorous (Wirtz, 2012), however, diet is not a trait in itself
but rather a function of the feeding traits.

### 4.2  Taxonomic clustering of traits

The assumption that traits are conserved within taxonomic branches may not always hold. A
large part of the diversity of pelagic copepods has only briefly been described in the literature,
and little is known about the biology (Razouls et al., 2005-2016). Deeming a whole family to
carry a certain trait therefore often means extrapolating from a few well known species to
many rare species. While this may be reasonable for strongly conserved traits like myelination
of the nervous system, for feeding mode and spawning strategy the appropriateness is less
clear. Spawning strategy, for example, seems to be homogenous across most orders and
families, yet in some calanoid families, such as *Aetideae*, both free-spawners and sac-
spawners are found. Sometimes heterogeneity is observed even within genera: while the
species *Euaugaptilus magnus* was found to carry its eggs, all other observed species in that
genus are free-spawners (Mauchline, 1998). Our data on spawning strategy largely stems
from Boxshall and Halsey (2004) who defined spawning strategy family-wise but noted in
several cases that the assumption was not certain. We included these remarks in the comments
of the spawning strategy table.

### 4.3  Variance in quantitative traits

Quantitative traits are subject to measurement errors that may be significant, especially for
traits that are difficult to measure or depend on parameter estimates, such as physiological
rates (Kiørboe and Hirst, 2014). Where possible, we accounted for measurement errors by
reporting standard deviations. However, in many cases this information was either not
available, or it was not retrievable with a feasible effort.
Furthermore, most important quantitative traits are strongly modulated by the environment
(Kattge et al., 2011a). For example, we found a substantial intraspecific variation of adult
body size in *Calanus finmarchicus* across the North Atlantic. Such variation is a consequence
of genetic variation and phenotypic plasticity and may optimize fitness in response to biotic



and abiotic environmental conditions. It may be interesting to study on its own, however, if
not properly quantified it introduces significant uncertainty to the data: point estimates from
particular individuals and locations that happen to be in the dataset may be an unrealistic
representation of the species (Albert et al., 2010). We tried to account for this problem by
including multiple trait measurements per species or averages over several measurements:
however, for many species no more than one value could be found. The large investment
required to measure copepod traits in the open ocean makes it difficult to overcome this
limitation in the near future.
**5    Data availability**
The data can be downloaded from PANGAEA, doi:10.1594/PANGAEA.862968.
**6    Conclusions**
We produced a database on key functional traits of marine copepods that may currently be
unique in its trait coverage and taxonomic breadth, enriching the field of trait-based
zooplankton ecology. It may be used to obtain an overview over correlations between traits, to
investigate the taxonomic and spatiotemporal patterns of trait distributions in copepods (e.g.,
Brun et al., *submitted*), or to inform and validate trait-based marine ecosystem models.
However, due to environmental modulation of many quantitative traits and the limited data
availability, the database may not always provide robust estimates on the species level,
making more detailed comparisons difficult. A way to overcome this uncertainty may be to
investigate relationships between traits measured for the same individuals or groups of
individuals, where the trade-offs are acting. Flexible structures for trait databases which are
capable to store such information have been developed for plants (Kattge et al., 2011a) and
successfully implemented in comprehensive efforts maintained by the scientific community
(Kattge et al., 2011b). Learning from these experiences may lift the field of trait-based
plankton ecology to the next level.



## Appendix A: List of important pelagic families considered in figures

*Acartiidae, Aetideidae, Arietellidae, Augaptilidae, Calanidae, Candaciidae, Centropagidae,*
*Clausocalanidae, Diaixidae, Discoidae, Eucalanidae, Euchaetidae, Heterorhabdidae,*
*Lucicutiidae, Megacalanidae, Metridinidae, Nullosetigeridae, Paracalanidae, Phaennidae,*
*Pontellidae, Pseudodiaptomidae, Rhincalanidae, Scolecitrichidae, Spinocalanidae,*
*Stephidae, Subeucalanidae, Sulcanidae, Temoridae, Tharybidae, Tortanidae, Cyclopinidae,*
*Oithonidae, Monstrillidae, Corycaeidae, Lubbockiidae, Oncaeidae, Sapphirinidae,*
*Aegisthidae, Euterpinidae, Harpacticidae, Miraciidae, Tisbidae, Misophriidae, Monstrillidae,*
*Mormonillidae, Caligidae, Pseudocyclopidae, Peltidiidae, Platycopiidae*





## Appendix B: Estimation of $Q_{10}$ value

Physiological rates measured at different temperatures were assumed to be related through the
following law:
$$R_{T2} = R_{T1} * Q_{10}^{\frac{T2-T1}{10}} \tag{A1}$$

where R stands for respiration rate at different temperatures T. The $Q_{10}$ value is the factor by
which respiration rates increase when temperature is increased by 10°C. We estimated $Q_{10}$
from the data by transforming Eq. (A1) and fitting a linear regression. The regression
equation was
$$ln\left(\frac{R_{T2}}{R_{T1}}\right) = \frac{1}{10} ln Q_{10} * (T_2 - T_1) \tag{A2}$$

Reference rates $(R_{T1})$ and temperatures $(T_1)$ where defined species-wise as the record taken at
the minimum temperature and used to calculate differences/ratios for all observations, which
were then used in the regression. Based on this procedure we estimated a $Q_{10}$ value of 2.14
(adj. $R^2 = 0.53$, df = 465).



## Acknowledgements

We thank Mänu Brun for support in the data collection and Hans van Someren Greve for the beautiful copepod illustrations. Furthermore, we acknowledge the Villum foundation for support to the Centre for Ocean Life.



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



**Tables**
Table 1: Important references used in the database and their taxonomic and geographical foci;
a full list of references is given in the data tables

| *Reference* | *Trait(s)* | *Focal taxa* | *Focal region* |
|---|---|---|---|
| Benedetti *et al.* (2015) | Feeding mode | Abundant copepods | Mediterranean Sea |
| Boxshall and Halsey (2004) | Spawning strategy | *Calanoida* | Global |
| Conway *et al.* (2003) | Body size | Copepods | Southwestern Indian Ocean |
| Conway (2006) | Body size | Common planktonic copepods | North Atlantic |
| Conway (2012) | Body size, spawning strategy | Copepods | Southern Britain |
| Hirst and Kiørboe (2014) | Body size | Copepods | Global |
| Ikeda *et al.* (2007) | Respiration rate | Marine pelagic copepods | Global |
| Kiørboe and Hirst (2014) | Clearance rate, ingestion rate, growth rate, respiration rate | Marine pelagic copepods | Global |
| Lenz (2012) | Myelination | *Calanoida* | Global |
| Mauchline (1998) | Egg size, clutch size, fecundity, hibernation, resting eggs, generations | *Calanoida* | Global |
| Neuheimer *et al.* (2016) | Egg size | Copepods | Global |



| Razouls *et al.* (2005-2016) | Body size | Marine planktonic copepods | Global |
|---|---|---|---|
| Walter and Boxshall (2016) | Taxonomy | Copepods | Global |




Table 2: Feeding modes included in the database and their categorization by feeding activity

| Feeding activity | Feeding modes |
|---|---|
| Passive | Ambush feeding |
|  | Particle feeding |
| Active | Feeding currents |
|  | Cruise feeding |
| Mixed | Combination of active and passive modes |
| Other | Parasitic |




Table 3: Abbreviations used for the classifications of life stage and sex in the database

| Abbreviation | Definition |
| --- | --- |
| NI, NII, NIII, NIV, NV | Naupliar stages 1-5 |
| N | Nauplius, no information about stage |
| CI, CII, CIII, CIV, CV | Copepodite stages 1-5 |
| C | Copepodite, no information about stage |
| A | Adult (copepodite stage 6), no information about sex |
| F | Adult female |
| M | Adult male |




## Figure captions


Figure 1: Copepod traits included in the database, arranged according to the framework of
Litchman et al. (2013). The vertical axis groups traits by trait type and the horizontal axis by
ecological function. Body size (bold) transcends several functions.
Figure 2: Trait-wise data coverage for taxonomic families of marine copepods. Top: number
of database records per trait; Left: Taxonomic tree of important families weighted by number
of species, including illustrations of type species for the dominant orders. Illustrated species
are (from top to bottom) *Calanus finmarchicus*, *Metridia longa*, *Oithona nana*, *Microsetella*
*norvegica*, *Monstrilla helgolandica*, *Oncaea borealis*, and *Caligus elongatus*, representing
orders according to their color code; Right: Table indicating the fraction of species for which
data was collected per family and trait. Note that since some traits are taxonomically
clustered, few records for higher order taxa may suffice to describe the entire diversity. *We
likely covered the vast majority of hibernating species and species with resting eggs that have
been reported in the literature. Yet, future discoveries may expand this list.
Figure 3: Variation of body size in marine copepods as a function of taxonomy, life stage and
location. Panel (a) shows boxplots of total body length for the most important families
covered. Thick lines on boxplots illustrate median, boxes represent the interquartile ranges
and whiskers encompass the 95% confidence intervals. Total length of *Calanus finmarchicus*
as a function of copepodite stage in two different areas is shown in panel (b). For males and
females mean values are shown as solid lines and mean ± standard deviation are shown as
transparent polygons. Distribution of female prosome length of *C. finmarchicus* in the North
Atlantic is shown in panel (c).
Figure 4: Physiological traits of adult copepods grouped by family, and frequency of life stage
information available for the records. Family-wise boxplots for clearance rate (a), ingestion
rate (c), growth rate (e), and respiration rate (f). Illustrated rate values are per mg carbon and
corrected to 15 °C. Thick lines on boxplots illustrate median, boxes represent the interquartile
ranges and whiskers encompass the 95% confidence intervals. Barplots in panels on the right
(b, d, g) indicate frequency distribution of life stage levels for the traits reported.
Figure 5: Taxonomic distribution of feeding modes in the most important families of marine
planktonic copepods. Distinguished are active feeders (blue), mixed feeders (orange), passive
feeders (green), and parasites (pink). Taxa for which no information was available are shown





in grey. Colors are mixed according to the fractions of trait carrying species in each
taxonomic group.
Figure 6: Reproductive traits grouped by family: Family-wise boxplots for egg diameter
including converted values from µg carbon (a), generations per year (b), clutch size (c), and
fecundity (d). Thick lines on boxplots illustrate median, boxes represent the interquartile
ranges and whiskers encompass the 95% confidence intervals.
Figure 7: Taxonomic distribution of binary traits in the most important families of marine
planktonic copepods. Fraction of trait carrying species is illustrated down to the family level
for myelination (a), spawning strategy (b), hibernation (c), and resting eggs (d). Families in
which the trait is present in at least one species are labelled.



**Figures**
Fig. 1

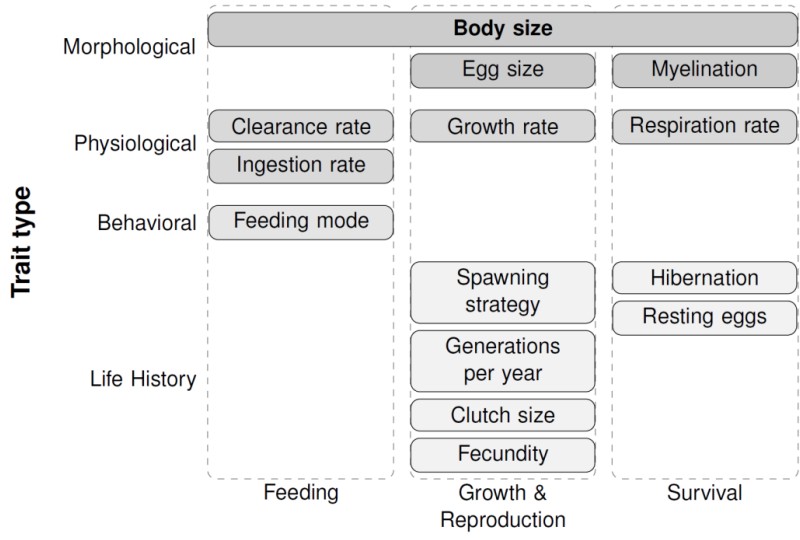






Fig. 2





Fig. 3

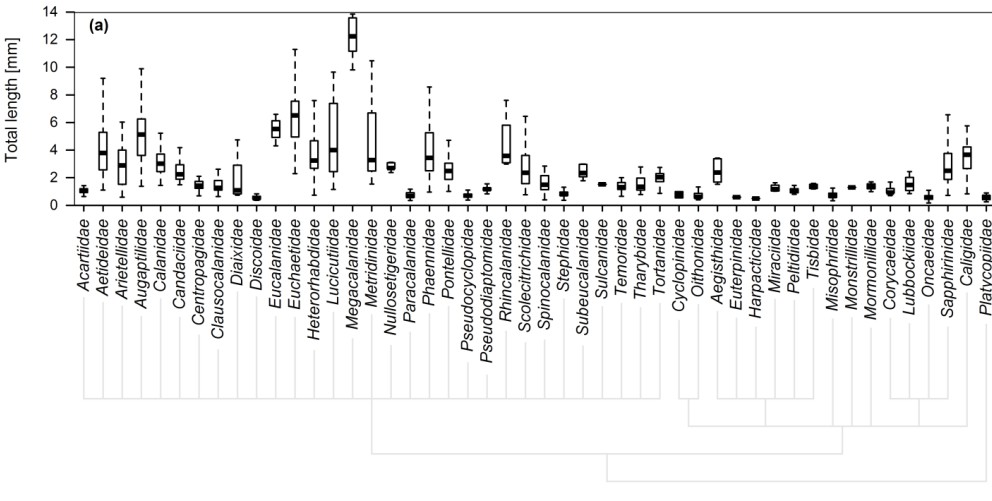

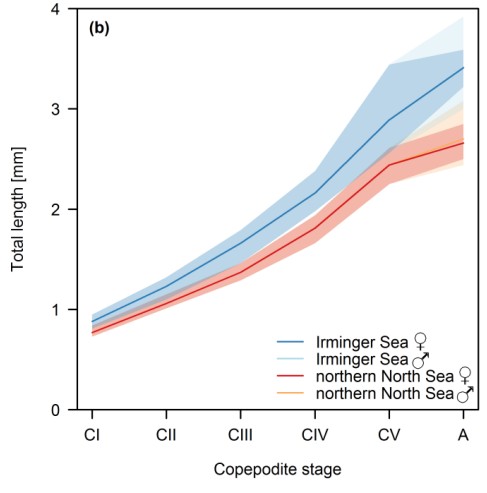

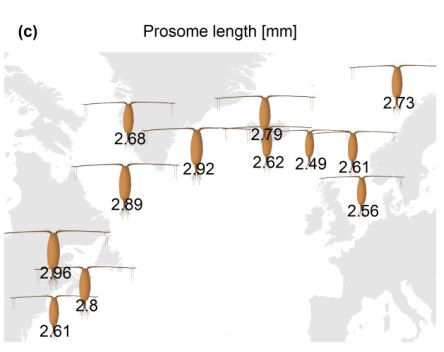





Fig. 4

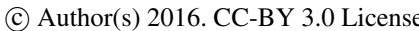





Fig. 5

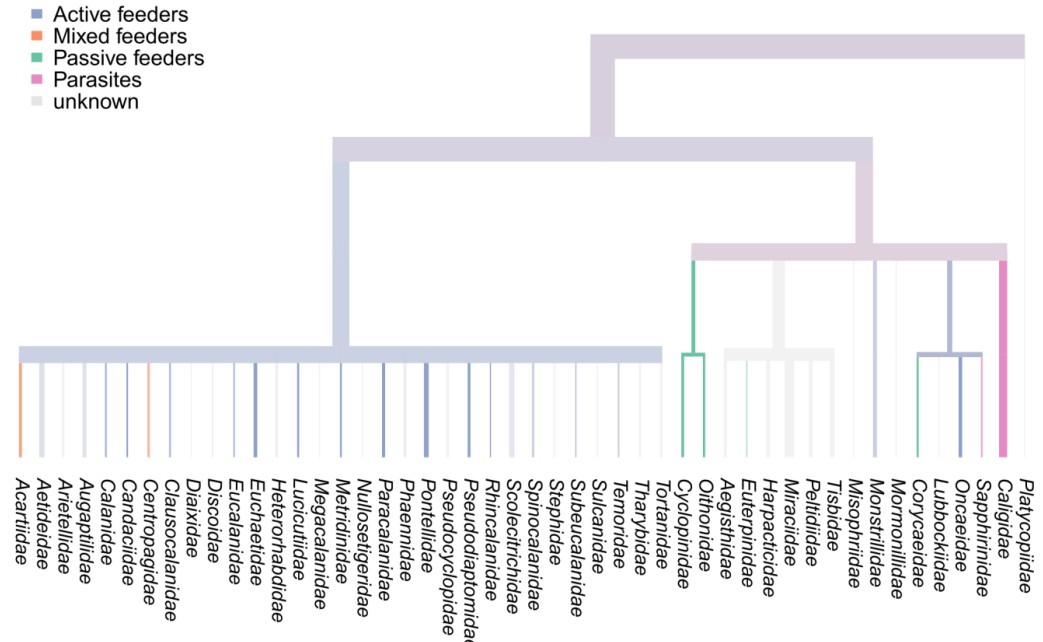





Fig. 6

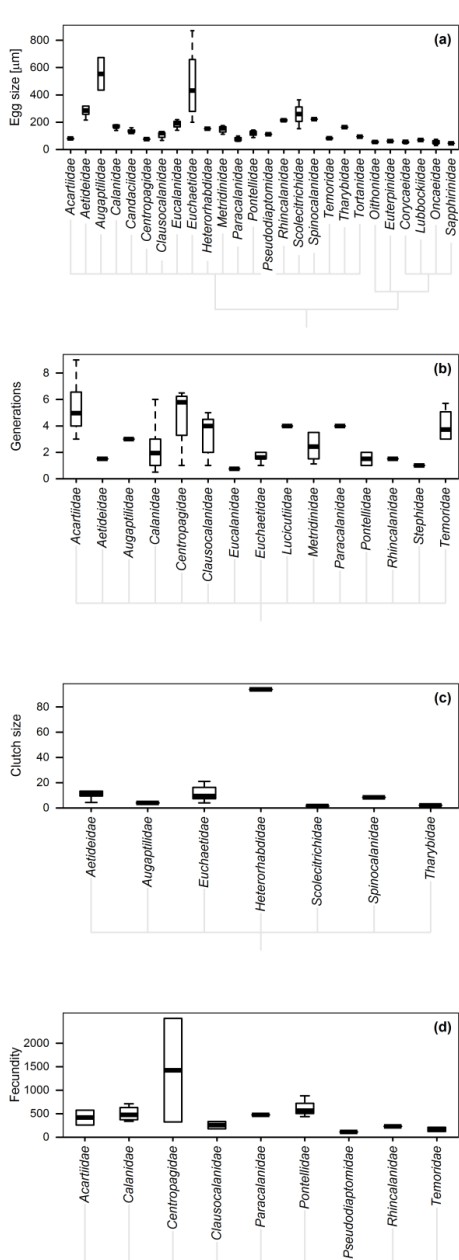





Fig. 7


