# Peer review of "Searth System Discussion Science Solutions Data"

_Earth System Science Data, 2016_

## Referee Comment (RC1) · Anonymous Referee #1 · 14 Sep 2016

Manuscript Reviews Manuscript for Earth Syst. Sci. Journal Data Discuss., doi:10.5194/essd-2016-30, 2016 Title: A trait database for marine copepods by Philipp Brun, Mark R. Payne and Thomas Kiørboe

The authors build a very interesting dataset and wrote a very interesting article. The objective was to construct, structure and make available for the scientist community the first large dataset about copepod life traits that they tried to make it as exhaustive as possible. (based on current knowledges and data availability. To do so, the authors first explained clearly the definition of life traits and why they are important to study. Recently, the trait-based has started to be more and more popular; marine ecology. This new approach used to better understand marine ecosystem functioning requires urgently new kind of data to be perform, preferably in a large amount and over wide ranges of taxons, geographical area, phenological stages, . . . So far, those information have been available but scattered in different labs, teams or research. They also have

been set in specific purpose, so build with a specific format, included a few numbers of life history traits. The authors pinpointed this lack and proposed to remedy that issue proposing a dataset that has been constructed with the requirements like accessibility, homogenisation, exclusivity, large amount number of data as constraints for their roadmap. Authors compiled traits information on marine pelagic copepods, from published literature and from experts, and organized the data into a structured database. They collected up to 11 9345 records for 14 functional traits. Particular attention was given to body size, feeding mode, egg size, spawning strategy, respiration rate and myelination. Authors proposed their database to be used to investigate relationships between traits, to produce trait biogeographies, or to inform and validate trait-based marine ecosystem models.

On the whole this paper is well written and makes a very useful contribution to the field. The methodology used sounds adapted. From my point of view, the following objectives are highly relevant and I like the way they were addressed. The methodology used is, as far as I can judge, well conceived, and well executed. The resulting dataset can be easily accessed and Such data compilation exercise is crucial to allow 'traits-based modelling and approaches to be successfully and fully used. That kind of new studies combined with a relevant and reliable dataset are excepted to provide solid results in the near future for our understanding of marine ecosystem dynamics in a changing world. However, I am not a "traits-based modelling" specialist and hence the editors might want to seek additional advice from colleagues, who will be far more competent in matters linked to relevance of the dataset for future research. Nevertheless, despite a complete and solid methodology part, some information are not enough developed in the present manuscript about the strengths and relevance of "traits-base modelling" compare to other classical approaches. I think those information could be useful for non-specialist to understand the objectives of this work.

Please note that I used the line numbering from the .pdf file version provided by the journal.

**GENERAL COMMENTS**

In introduction, authors explain clearly and in a concise manner what is the definition of "traits" for organisms, what are their roles, and how the "traits-based approach" differs from more classical and historical approaches which considers, typically, links between environmental factors and species or group of species. Trait-based approach studies diversity and describes the structure and function of communities or ecosystems among key traits. Nevertheless, at this point I think an explanation is missing about what the "trait-based" approach can bring for the understanding of the ecosystem that "classical-historical approaches" can not. If I am convinced about the novelty and the legitimacy of using that new approach, I don't see what are the expectations compare to those that could been issued from a classical approach, in the introduction. The authors write lines 29-30 "...feeding, survival and reproduction. Functional traits determine the outcome of one or several of those missions..."; but I do think that results for those missions can be determined by less mechanistic approach like the impact of environmental factors, the quality and abundance of resources. I would like to understand why the authors have chosen "trait-based study" among others, apart from the innovation part. This question may look "out of the subject" I guess for people who are familiar with "traits-based" approaches but for scientist who are not, it would be helpful to have some further explanations, without going to far in the details.

**SPECIFIC COMMENTS**

About trait information in Part 2,2 of the manuscript, authors reported that it is not always straightforward to assign a "value" of trait to an organism because of lack in taxonomic resolution or because the "behaviour" displayed by the individual can fall in different category for one trait (like feeding category). Both quantitative and qualitative trait can been difficult to allocate. Suggestion: would it be possible to provide a kind of level of confidence? That could help the user to have an idea about the reliability of the assignation and so to decide whether or not the data can be used for her/his purpose.

Figure 2 I am not sure to understand the color circle code. I understand the colorbar, gray means no data, and then from 0 to 100 the % of coverage for taxonomic families of marine copepods. But what about circles that have only their perimeter in color, and a different color inside? For example: Calanidae second column ? Is it about that as some traits are taxonomically clustered, few records for higher order taxa may suffice to describe the entire diversity?

SPECIFIC QUESTIONS

Is the article itself appropriate to support the publication of a data set?

The article itself is appropriate to support the publication of the dataset. The quality of the data is sufficient enough, and the size of the data is expected to grow in the near future. The dataset can be easily downloaded from the given identifier, nevertheless I would recommend the author to provide a direct HTML link inside the manuscript. Authors provided details about the ranges of the data in terms of : type of traits, taxonomic distribution, organism length, biogeography, in the result part of the manuscript. Furthermore, authors acknowledge that variance in quantitative traits are subject to measurement errors that may be significant, so where possible, they accounted for measurement errors by reporting standard deviations.

Is the data set significant – unique, useful, and complete?

The dataset is significant enough to be useful and usable although some would qualify it not big enough or to specialized on copepods zooplankton. The database is expected to grow, and there is room for work using just copepods organisms data regarding to their importance in the marine ecosystem.

Is the data set itself of high quality?

It self the current dataset is conveniently down-loadable and usable. There is no need of a proprietary software to access and manipulate the data since you have collected them. Furthermore, the setting of the data inside the dataset is clear, easy to understand and to manipulate for a personal research. The article itself is quick and concise, its structure allow to clearly understand the goal and the specificity of the dataset. The mathematical part is short and correct. Expect Fig. 2 (see my comment earlier), all the figures and table are clear and ready to be published.

To summarize, I consider the dataset understandable and practical enough for a personal use.
* * *

---

## Author Comment (AC1) · 14 Sep 2016

We thank Referee # 1 for taking the time to read the manuscript and for the valuable and constructive feedback. From the comments we identify four key issues that we intend to address in the revised manuscript as we specify below.

1. Elaborate more on advantages of the trait-based approach compared to traditional approaches in the introduction.

We will include an in depth description of the usefulness of the trait-based approach as a meaningful way to simplify complex, species-rich ecosystems that facilitates mechanistic analyses of key ecosystem processes.

2. Provide a level of confidence for trait estimates.

This is a good point. For continuous traits, we already report uncertainty values and ranges, where the information is available, but we do not do this in a systematic way

for categorical traits. Where we have information that classifications are uncertain, we mentioned this in the comment columns. However, we agree that it would be more useful to have an additional column in the tables that flags these observations and allows users to easily filter for them. We will add such a column to the tables covering categorical traits.

3. Describe orange rings in the legend of Figure 2.

In Figure 2 we added bright orange rings to the data coverage information for hibernation and resting eggs to indicate that information for these traits likely is near complete and covers all marine copepods. We only report a few dozen species that hibernate or produce resting eggs, but we likely considered all information available in the literature. We therefore assume that, at least in terms of biomass, the reported taxa represent the large majority of hibernating and resting egg producing copepods. There may be hibernating and resting egg producing taxa which have not been observed yet, and therefore are not covered in our database, but if this is the case they would probably be rare and insignificant in terms of biomass. We will make this point clearer in the figure legend.

4. Add a HTML link to the data in the manuscript.

As far as we understand, this will be included in the typeset manuscript.

―――――――――――――――――――――

---

## Referee Comment (RC2) · F. Maps (Referee) · 16 Nov 2016

The authors present a comprehensive database of both qualitative and quantitative morphological, physiological and behavioural characteristics of copepod species, as well as critical information on their life-cycle strategies. They reviewed the existing literature, and asked experts in the field as well, to gather an impressive and unprecedented amount of information that they organized under the theoretical and practical framework of "trait-based ecology". This approach is well established and has proven its worth in terrestrial plant ecology, and its gaining momentum in marine ecology as well for obvious reasons of efficiency and flexibility. I think their work is timely, sound, very useful and well presented. I really appreciated the conciseness of the paper, given the daunting task that had to be accomplished. I hope though that the authors will agree with me that their work would benefit from the few suggestion listed below (Line numbers refer to the PDF).

[Figure]

Specific comments

L31-35: I appreciate the concise definition of a trait provided here, even though what a "trait" exactly is is still open for debate (the authors acknowledge that in their discussion). L58: one such "trait" that I would question is specifically the number of generations per year. I understand the motivation for that (fast growing vs slow growing species, R vs K strategies, potential for invasiveness, etc.) but this supposedly quantitative trait is extremely dependant on the environmental conditions (temperature, first, but also food conditions – quantity, quality, etc.). I would argue that it almost contradicts the very definition of a trait given above in the introduction. Moreover, I was surprised not to find development rate as a trait! I think this is absolutely key to any aspects of copepods ecology. Actually, the main trait this study focuses on, body size, results essentially from the trade-off between growth (accumulation of matter) and development (differentiation of tissues) according to the abundant and insightful work of some of the authors. And the number of generation is also essentially the expression of the development speed of the species. I would like very much the authors to explain why they decided to leave out development in their thorough compilation of copepod traits.

L117: About units. Unless I'm mistaken, units are not indicated in the spreadsheet. This should be corrected since it could lead to some errors by future users of the database.

L225: here specifically and elsewhere in the Results section, it would be interesting to note whether the taxon with the most variability is also the one with the most observations reported. Calanus species for instance are undoubtedly the most studied group, and hence it comes at no surprise that a wide range of values have been reported for a wide range of experimental and environmental conditions, thus certainly increasing the interquartile range (and maybe the mean).

L247: this is a surprisingly low value... the large C. hyperboreus can definitely produce clutches of > 100 eggs during peak production. It's really just a remark since this

database will evolve and be complete in time... Actually, it shows that this work could also be useful to gauge whether a simple unpublished dataset could be worth the "trouble" of publishing! If the authors have thought of an outlet to publish verified but unpublished copepod-related data, it would be great to know...

L337: This is a suggestion: I think this impressive work granted the authors the right to share with the research community their thoughts on the "best practices" that should be adopted sooner than later by data-producing marine ecologists. How shameful it is to realize that such obvious information as development stage is rarely available within publications! It would be extremely valuable to include a paragraph, in the form of a few guidelines (minimum metadata to include, format of spreadsheets, of supplementary material, etc.), to enhance our ability to complement rapidly and efficiently this database.

Finally, apologies for the late review...